# Population Genetic Structure Is Unrelated to Shell Shape, Thickness and Organic Content in European Populations of the Soft-Shell Clam *Mya Arenaria*

**DOI:** 10.3390/genes11030298

**Published:** 2020-03-11

**Authors:** Michele De Noia, Luca Telesca, David L. J. Vendrami, Hatice K. Gokalp, Grégory Charrier, Elizabeth M. Harper, Joseph I. Hoffman

**Affiliations:** 1Department of Animal Behavior, University of Bielefeld, Postfach 100131, 33615 Bielefeld, Germany; Michele.DeNoia@glasgow.ac.uk (M.D.N.); david.vendrami@student.unife.it (D.L.J.V.); haticekubra.gokalp@stu.uskudar.edu.tr (H.K.G.); 2Department of Earth Sciences, University of Cambridge, Downing Street, Cambridge CB2 3EQ, UK; lt401@cam.ac.uk (L.T.); emh21@cam.ac.uk (E.M.H.); 3British Antarctic Survey, High Cross, Madingley Road, Cambridge CB3 OET, UK; 4Institute of Biodiversity, Animal Health & Comparative Medicine, College of Medical, Veterinary & Life Sciences, University of Glasgow, Glasgow G12 8QQ, UK; 5University Brest, CNRS, IRD, Ifremer, LEMAR, F-29280 Plouzané, France

**Keywords:** *Mya arenaria*, soft-shell clam, microsatellite, population genetic structure, phenotypic plasticity, shell morphometrics

## Abstract

The soft-shell clam *Mya arenaria* is one of the most ancient invaders of European coasts and is present in many coastal ecosystems, yet little is known about its genetic structure in Europe. We collected 266 samples spanning a latitudinal cline from the Mediterranean to the North Sea and genotyped them at 12 microsatellite loci. In parallel, geometric morphometric analysis of shell outlines was used to test for associations between shell shape, latitude and genotype, and for a selection of shells we measured the thickness and organic content of the granular prismatic (PR), the crossed-lamellar (CL) and the complex crossed-lamellar (CCL) layers. Strong population structure was detected, with Bayesian cluster analysis identifying four groups located in the Mediterranean, Celtic Sea, along the continental coast of the North Sea and in Scotland. Multivariate analysis of shell shape uncovered a significant effect of collection site but no associations with any other variables. Shell thickness did not vary significantly with either latitude or genotype, although PR thickness and calcification were positively associated with latitude, while CCL thickness showed a negative association. Our study provides new insights into the population structure of this species and sheds light on factors influencing shell shape, thickness and microstructure.

## 1. Introduction

The soft-shell clam *Mya arenaria* Linnaeus, 1758, is a marine bivalve that occurs in numerous intertidal infaunal communities across Europe and North America. While this species has a relatively high dispersal potential during the planktonic larval stage and as juveniles [1], the contemporary geographic distribution of *M. arenaria* appears to have been largely influenced by human-mediated translocations. Although deliberate introductions have been documented in the literature [2], the majority of introductions were probably unintentional, occurring either as a byproduct of oyster transplants [3] or via shipping, as this species can survive for extended periods in ballast water [4,5].

Present in Europe during the Pliocene [6], the soft-shell clam is believed to have been extirpated from this continent during the Pleistocene glaciations [7]. Radiocarbon dating of shells found along the coast of the North Sea suggests that *M. arenaria* was subsequently reintroduced into Europe by the Vikings during the 13th to 15th centuries [8,9], making it one of the oldest marine invaders of European coasts [10]. Following recolonization from the North Sea and probable further man-mediated introductions, the soft-shell clam can nowadays be found around most of the continent, including in the Mediterranean Sea [11,12], on the Iberian Peninsula [13], along the Atlantic coast of France [14], around the British Isles [15,16] and in the North Sea [1,17,18].

Despite *M. arenaria* being an important species in many European coastal ecosystems, little is known about its population genetic structure, especially when compared to other European shellfish species [19,20,21,22,23]. In addition to two studies of *M. arenaria* from North America, both of which included a single European population [24,25], only three studies to our knowledge have focused on the population genetics of the soft-shell clam in Europe [5,26,27]. In all cases, little in the way of population genetic structure could be detected and genetic diversity appeared to be low. However, these studies were exclusively based on either allozymes or mitochondrial DNA (mtDNA), which provide limited power to detect genetic differentiation, especially when population structure is shallow.

More recently, microsatellite markers have been developed for the soft-shell clam, which, in contrast to the studies described above, have uncovered relatively strong population genetic structure and higher levels of genetic diversity in North America [25,27,28]. One of these studies [27] also generated microsatellite data for four European *M. arenaria* populations, although these were locally restricted to the Southern coast of Ireland, North Wales and the Netherlands. This study found clear evidence for genetic differentiation at microsatellites but not mtDNA [27], implying that population genetic structure may be present over a broader scale in Europe.

Characterizing population genetic structure not only provides information on patterns of gene flow and genetic drift, but it may also shed light on the extent to which morphological or other phenotypic traits are plastic versus under genetic control [29,30]. The underlying logic of this approach is to relate phenotypic differences among populations to the underlying genetic structure. On the one hand, the presence of pronounced phenotypic differences among genetically indistinguishable populations may be suggestive of a strong influence of the environment on the traits in question. On the other hand, coincident patterns of genetic and phenotypic variation point towards a possible role of genetics. Shell morphological traits of marine invertebrates provide an ideal test bed for investigating the relative contributions of genes and the environment towards phenotypic trait variation, as protocols have already been established for quantifying shell morphological and microstructural variation [31,32,33] and it has been suggested that phenotypic plasticity may play an important role in determining individual variation in shell features of many marine organisms including limpets [34], sea snails [35] and scallops [36].

One obvious shell morphological trait where great variation among individuals can be observed is shape. Shell shape variation has been studied in a variety of molluscan species and it appears to be significantly influenced by both environmental and genetic effects, as well as their interaction [37,38,39]. Abiotic factors associated with shape variation include water temperature and acidity [40], while an effect of latitude has also been reported [41], probably due to changes in environmental conditions linked to latitude. Moreover, biotic variables such as predation have also been demonstrated to be important in determining individual variation of shell shape [42]. However, little is known about the specific causes of shell shape variation in *M. arenaria*. While Swan [43] and Emerson [44] have provided evidence for a potential role of substrate typology and water flow, to our knowledge no study has investigated the potential effect of genetics on shell shape variation.

The shell of *M. arenaria* is composed of four main layers. These are the outermost periostracum, the granular prismatic (PR), the crossed-lamellar (CL) and the complex crossed-lamellar (CCL) layers [45] (Figure 1 and Appendix A). While the periostracum consists of sclerotized proteins, the other layers are mainly composed of aragonite (CaCO_3_) crystals and inter-crystalline biomineral organic matrix. Differences in the energetic cost of producing and maintaining different shell structures and components [46,47] as well as geographical variation in physical and biotic stressors that are likely to exercise a selective pressure on shell morphology, are expected to influence variation in shell microstructural composition and thickness [48,49]. The fact that *M. arenaria* is widespread and locally abundant, combined with the availability of recently developed microsatellites [25,28], makes this species a good candidate for investigating the relative contributions of genetic and geographical factors towards variation in shell morphological traits.

Here, we collected a total of 266 *M. arenaria* samples from nine locations around the coastline of Europe, spanning a latitudinal cline from Lisbon to St. Andrews (Figure 2). Genetic data for 12 microsatellite loci were generated and analyzed in combination with data on shell shape, total thickness and the thickness of the PR, CL and CCL layers. For a subset of samples originating from the extreme northern and southern sampling sites, the organic content of these layers was also measured. This allowed us to characterize the pan-European population genetic structure of this species and to test for effects of genotype and latitude on key shell characteristics.

## 2. Materials and Methods

### 2.1. Sample Collection and DNA Extraction

Between 15 and 40 *M. arenaria* specimens of wild origin were collected from the eulittoral zone of each of nine locations within Europe (Figure 2, Table 1). The length of the shell of every individual was measured with digital calipers (0.01 mm precision) and used as a within-population proxy for age [50]. The shells were retained and processed to determine shell shape as well as to quantify shell thickness and organic content as described subsequently. Approximately 1 cm^3^ of tissue, either adductor muscle or mantle, was taken from each individual and stored in 95% ethanol at −20 °C for genetic analysis. Whole genomic DNA was extracted following an adapted phenol-chloroform protocol [51].

### 2.2. Microsatellite Genotyping

All of the samples were genotyped at 12 previously characterized microsatellite loci [25,28]. These were polymerase chain reaction (PCR) amplified in four separate multiplexed reactions using a Type It Kit (Qiagen) with the following PCR profile: one cycle of 5 min at 95 °C; 30 cycles of 30 s at 94 °C, 90 s at the specified annealing temperature (T_a_) and 30 s at 72 °C; and final elongation step of 15 min at 60 °C (see Appendix A for T_a_). Fluorescently labeled PCR products were resolved by electrophoresis on an ABI 3730xl capillary sequencer and allele sizes were scored by three independent observers using GeneMarker v. 2.6.2 (Softgenetics^®^). Samples that failed to genotype at four or more loci were excluded from subsequent analyses.

### 2.3. Genetic Summary Statistics

The R package Pegas v. 0.12 [52] was used to test for deviations from Hardy-Weinberg equilibrium at each locus using 1000 Monte Carlo replicates, while Genepop on the Web [53,54] was used to check for deviations from linkage equilibrium using default parameters. The resulting *p*-values were corrected for the table-wide false discovery rate (FDR) according to the procedure described by Benjamini & Hochberg [55]. Next, the R package diveRsity v. 1.9.90 [56] was used to calculate the number of alleles (*N*_a_), allelic richness (*A*_r_), observed heterozygosity (*H*_o_) and expected heterozygosity (*H*_e_). Finally, we calculated standardized multilocus heterozygosity (sMLH) for each individual using the R package inbreedR v. 0.3.2 [57].

### 2.4. Analysis of Population Structure

To test for the presence of population structure, we implemented a number of different approaches. First, we calculated pairwise *F*_st_ values using Arlequin v. 3.5.2.2 [58], where statistical significance was determined based on 1000 permutations of the dataset. The resulting values were then used together with measures of shortest coastline distances among populations to test for the presence of isolation-by-distance by implementing a Mantel test. Second, we conducted a principal component analysis (PCA) of the microsatellite dataset using the R package adegenet v. 2.1.1 [59,60]. Third, we used Structure v. 2.3.3 [61] to carry out a Bayesian cluster analysis. Structure uses an iterative approach to group individuals into *K* groups by dividing the dataset in such a way that maximizes Hardy-Weinberg and linkage equilibria within the resulting groups. Each individual is then attributed a group membership value (*Q*) that varies from 0 to 1, with the latter indicating full group membership. We ran five simulations for each value of *K* between one and 10. We set the burn-in period and Markov chain Monte Carlo repetitions to 10^5^ and 10^6^, respectively. The most likely number of genetic groups was evaluated using the maximal average value of Ln *P* (*D*), a model choice criterion that estimates the posterior probability of the data, and the Δ*K* procedure described by Evanno et al. [62]. 

### 2.5. Elliptic Fourier Analysis of Shell Outlines

A geometric morphometric approach [63] based on elliptic Fourier analysis (EFA) of the shell outlines [64] was used to describe shape variation both within and among populations. Geometric information was extracted from the shell outlines and described as periodic function [32] through decomposition into the harmonic sum of progressively simplified trigonometric functions [65]. Low-frequency harmonics were used to approximate coarse-scale variation in the outlines, whereas higher-frequency harmonics captured fine-scale variation [32].

Outlines of orthogonal lateral and anterior views of the left shell valves (total *n* = 262; Figure 1b) were digitized and used as input data. The outlines for both views were processed independently, geometrically aligned and later combined for analysis following the protocol of Telesca et al. [31]. We then implemented an EFA on the resulting coordinates from shapes invariant to outline size, translation and rotation. After calibration, we chose seven harmonics to retain 99% of the cumulative harmonic power [66] (Appendix A). Four coefficients per harmonic (28 descriptors per view) were then extracted for each shell outline and used as variables quantifying geometric information [31,67].

A PCA was performed on the matrix of harmonic coefficients to characterize shell shape variation among individuals. The first three principal components (PCs), capturing 90.1% of the total shape variance, were used as new shape variables and analyzed using multivariate analysis of variance (MANOVA) to test for significant effects of location of origin and shell length (size) on shape variances. To visualize shell outline differences at the extremes of the morphospace, we generated deformation grids and iso-deformation lines through mathematical formalization of thin plate spline (TPS) analysis [68]. Shell morphometric analyses were carried out using Momocs v1.2.9 [32].

Generalized linear mixed models (GLMMs) were then used to explain variation in shell shape with respect to a number of predictors. Specifically, the scores of the first three shape PCs were modelled as a function of standardized latitude, the first two genetic PCs, sMLH, shape (a categorical variable with three levels: shape PC1, shape PC2, shape PC3) and their two-way interactions. Shell length was also included in the model to control for possible effects of within-population size variation, and collection site was fitted as random effect. Scores from the first three shape PCs were used as response variables within the same model to account for the interdependence of multiple shape variables that simultaneously describe variation in shell outlines as a whole [31]. This model was then optimized by rejecting non-significant interaction terms and factors that minimized the AICc value (Appendix A). The final model was then of the form:(1)$$\begin{array}{ll} ShellShape_{ijk} & =Latitude_{ik}+Length_{ik}+gPC1_{ik}+gPC2_{ik}+sMLH_{ik}+ShapePC_{j}\\ & +Length_{ik}\times Layer_{j}+Site_{ij}+\varepsilon_{ijk}\\ & \;Site_{ij}\;\sim \;N\left(0;\;\sigma_{Site}^{2}\right)\\ & \;\varepsilon_{ijk}\sim \;N\left(0;\sigma^{2}\right) \end{array}$$
where ShellShapes*_ijk_* is the *k*th thickness observation from PC*_j_*(*j* = PC1, PC2, PC3) and site *i* (*i* = 1, …, 9), Site*_ij_* is the random intercept and ε*_ijk_* is the error, which are assumed to be normally distributed with an expectation of zero and variances $\sigma_{Site}^{2}$ and $\sigma^{2}$ respectively. Mean effect sizes of the predictor variables were estimated from the optimal model fitted on standardized variables [69]. Due to the difficulty of reliably estimating *p*-values in mixed models [70], we considered as significant any effect whose 95% confidence interval (CI) did not overlap zero. CIs were generated using a bias-corrected parametric bootstrap approach with 10,000 iterations of the data.

### 2.6. Analysis of Shell Thickness 

A total of 167 shells were characterized in terms of both total thickness and the thickness of the three individual layers (PR, CL and CCL). The left shell valves were set in polyester resin (Kleer-Set FF, MetPrep, Coventry, UK) blocks and sliced longitudinally along their axis of maximum growth (Figure 1c) using a diamond saw. Shell cross-sections were progressively polished with silicon carbide paper (grit size: from P800 to P2500) and diamond paste (grading: from 9 µm to 3 µm). Polished cross-sections were treated with Mutvei’s solution [33] to highlight different growth marks and microstructures within the shells (Figure 1c). The thickness of each shell layer was measured on photographs of polished sections that were acquired using a stereo-microscope (Leica M165 C equipped with a Leica DFC295 HD camera, Leica, Wetzlar, Germany). Since larger individuals had undergone environmental abrasion, which removed the granular prismatic layer near the umbo, we measured the thickness of the whole shell and individual layers at the midpoint along the shell cross-section. 

Next, we constructed two separate GLMMs to investigate how various predictor variables were associated with whole shell thickness and the thickness of the three individual layers. First, whole shell thickness was modelled as a function of standardized latitude, genetic PC1 and PC2, sMLH, shell length (which was included to control for possible effects of within-population size variation on layer thickness) and collection site, which was included as random effect. In the second model, the thicknesses of the three shell layers were combined into a single response variable and the following predictor variables were analyzed: standardized latitude, the first two genetic PCs, sMLH, shell layer (a categorical variable with three levels: PR, CL, CCL) and their two-way interactions. Shell length and collection site were also included, as in the model of whole shell thickness. The thickness of the three layers was analyzed within the same GLMM to simultaneously model common and divergent effects on each layer as well as to reduce the probability of type I error. Non-significant interaction terms were then rejected and only factors that minimized the AICc value (Appendix A) were retained in the final model. This was of the form:(2)$$\begin{array}{ll} Thickness_{ijk} & =Latitude_{ik}+Length_{ik}+gPC1_{ik}+gPC2_{ik}+sMLH_{ik}+Layer_{j}\\ & +Latitude_{ik}\times Layer_{j}+Length_{ik}\times Layer_{j}+Site_{ij}+\varepsilon_{ijk}\\ & \;Site_{ij}\;\sim \;N\left(0;\;\sigma_{Site}^{2}\right)\\ & \;\varepsilon_{ijk}\sim \;N\left(0;\sigma_{j}^{2}\times e^{2\delta \times gPC2_{ik}}\right) \end{array}$$
where *Thickness*_ijk_ is the *k*th thickness observation from *layer j* (*j* = PR, CL, CCL) and site *i* (*i* = 1, …, 9), *Site*_ij_ is the random intercept for layer *j*, which is assumed to be normally distributed with an expectation of zero and variance $\sigma_{Site}^{2}$. *ε*_ijk_ is the normally distributed error with a mean of zero and different variances per layer *j* changing exponentially with the variance covariate *gPC2*_ik_. This variance structure was chosen over others because it minimized the AICc value (Appendix A). Mean effect sizes of predictor variables were estimated following the same procedure described for the GLMM on shell shape (see Section 2.7).

### 2.7. Analysis of Organic Shell Content

We performed thermogravimetric analyses (TGA) to estimate the weight proportion (wt%) of organic matrix within the two dominant shell layers, PR and CCL. A random subsample of five *M. arenaria* specimens were selected from two populations Lisbon (LIS) and Saint Andrews (SAN) to test for differences in shell organic content between low and high latitudes. We removed the periostracum by sanding before isolating pieces of PR and CCL. Shell pieces were cleaned, air-dried and then finely ground. We tested 10 milligrams of this powdered shell with a thermogravimetric analyzer (TGA Q500, TA Instruments, New Castle, DE, USA). The samples were subjected to constant heating from ~25 °C to 700 °C at a linear rate of 10 °C min^−1^ under a dynamic nitrogen atmosphere and weight changes were recorded. The proportion of organic matrix (wt%) was then estimated as the proportion of weight loss during the thermal treatment between 150 °C and 550 °C (Appendix A). The wt% of organic matrix in the PR and CCL layers (*n* = 20) was then modeled as a function of collection site and shell layer to test for latitudinal differences in shell organic content. Pairwise contrasts with a standard Bonferroni correction were then used to test for differences in wt% between the layers both within and between sampling locations.

### 2.8. Data accessibility

The microsatellite data used to generate this study are publibly available from the Figshare repository (https://figshare.com) (DOI: 10.6084/m9.figshare.11949546).

## 3. Results

In order to investigate patterns of genetic and morphological variation in *M. arenaria*, we collected a total of 266 samples from nine locations spanning a European latitudinal cline. A total of 247 individuals were successfully genotyped at 12 microsatellite loci. Genetic variability was relatively high, with each locus carrying on average 17 alleles and observed heterozygosity averaging 0.69 (Table 1 and Appendix A). A number of significant deviations from Hardy-Weinberg equilibrium (HWE) were found after table-wide correction for the false discovery rate (Appendix A), but the vast majority of loci deviated from HWE in fewer than three populations so null alleles or genotyping errors are unlikely to be responsible. Nevertheless, locus Ma26 showed significant deviations from HWE in seven populations and we therefore took the conservative measure of excluding this marker from subsequent analyses.

### 3.1. Population Structure

Pairwise *F*_st_ values varied between 0.02 and 0.14 (Appendix A), with comparisons involving the Italian population (ITA) producing the highest overall values (mean = 0.12). The majority of *F*_st_ values were statistically significant after FDR correction (Appendix A). A significant pattern of isolation-by-distance was also obtained for the full dataset (Mantel’s *r* = 0.67, *p* < 0.05). Significance was lost after excluding ITA (Mantel’s *r* = 0.15, *p* = 0.3) but the overall relationship became significant again after both ITA and LIS were excluded (Mantel’s *r* = 0.52, *p* < 0.05). Results of the PCA confirmed the greater magnitude of differentiation of the Italian population by clearly resolving individuals from ITA as a separate cluster (Figure 3a), while the other eight populations were distributed more or less as a continuum along the second principal component axis.

Arguably the most powerful tests of population structure need not rely on knowledge of the populations from which individuals were sampled. We therefore used the program Structure to identify the most likely number of genetic groups (*K*) within our dataset and to derive group membership coefficients (*Q*) for each individual. The number of groups is often identified using the maximal value of Ln *P*(*D*), a model-choice criterion that estimates the posterior probability of the data. However, Ln *P*(*D*) often plateaus or continues to increase slightly after the true value of *K* has been reached [62]. Our data yielded just such a pattern, with Ln *P*(*D*) rising steeply until around *K* = 4 and then increasing gradually towards a peak at *K* = 9 (Figure 3b). We therefore calculated Δ*K*, an *ad hoc* statistic based on the second order rate of change of the likelihood function with respect to *K* that has been shown by Evanno et al. [62] to be effective at capturing the uppermost hierarchical level of population structure under most circumstances. Δ*K* peaked at three (Figure 3b), indicating support for three main genetic groups corresponding to (i) Italy; (ii) Brest, Plymouth and St. Andrews; and (iii) the remaining North Sea populations, Kiel and Lisbon (Figure 3c). Increasing *K* to four additionally resolved St. Andrews as a separate group (Figure 3d), while further increases in *K* did not appreciably change the overall pattern.

### 3.2. Shell Shape Variation

PCA performed on harmonic coefficients revealed marked variation of shell outlines among individuals in both lateral and anterior views. The first three PCs accounted for 90.1% of the shape variability and a scatterplot of PC1 versus PC2 revealed appreciable variation among the nine populations across the morphospace (Figure 4). Multivariate analysis of variance (MANOVA) indicated a significant influence of collection site on shell shape (Wilk’s *λ* = 0.51, approx. *F*_8, 252_ = 7.85, *p* < 0.01) but there was no effect of shell size (Wilk’s *λ* = 0.99, approx. *F*_1, 252_ = 0.60, *p* = 0.62).

Subsequently, we attempted to identify specific shell features making the greatest contributions towards shell shape variation through comparisons of outlines at the extremes of the morphospace (Appendix A). We furthermore decomposed shell shape variation according to the contributions of the first three PCs through visual inspection of shell outlines constructed for increasing values of each PC (Appendix A). Finally, we used the mean shape and TPS analyses to illustrate the main outline deformation required to pass from one extreme of the morphospace to the other (i.e., from population Plymouth (PLY) to SAN, Appendix A). We found that PLY was characterized by more elongated and deeper shells than SAN, which exhibited rounder shells and flatter anterior view profiles.

Finally, we constructed a generalized linear mixed model (GLMM) to explore the effects of latitude, body size and genotype (expressed as genetic PC1 and PC2 scores and individual standardized multilocus heterozygosity, sMLH) on shell shape variation, as described in the Materials and methods. Shell shape was not significantly associated with latitude or genotype (Figure 5a, Table 2), but a significant association was found between shell length (a proxy for age) and the third morphological PC (*p* < 0.01, effect size = 0.27, 95% CI = 0.07 to 0.45). Consequently, shell shape appears not to be influenced by any of the predictor variables with the possible exception of a weak effect of age.

### 3.3. Variation in Shell Thickness

In order to investigate how shell thickness may be related to latitude, shell length and genotype, we constructed two GLMMs, the first of whole shell thickness and the second of the thickness of the individual layers (see Materials and methods for details). Variation in whole shell thickness was not associated with any of the predictor variables, with the exception of shell length (Figure 5b, Appendix A), indicating an apparent absence of any latitudinal or genetic effects but a likely positive effect of age. In the GLMM of the thickness of the individual shell layers, we again detected a significant influence of shell length but no effect of genotype (Figure 5b, Table 2). However, this time, significant effects of latitude on the individual shell layers were detected, both in the form of a main effect of latitude and an interaction between latitude and shell layer. Specifically, the PR layer was proportionately thicker at higher latitudes and the CCL layer was proportionately thicker at lower latitudes, while the thickness of the CL layer did not appear to vary with latitude (Figure 5c).

### 3.4. Variation in Organic Content 

To investigate differential patterns of organic content deposition in the two main shell layers at different latitudes, we quantified the proportion of organic matrix in the PR and CCL layers for each of five individual shells taken from the two extremes of the latitudinal range (LIS and SAN). The PR layer was characterized by a significantly higher wt% of organic content in shells from warm temperate regions in comparison to cold temperate regions (mean difference = 0.53 wt%, *z* = 3.69, *p* < 0.01), indicative of increasingly calcified prismatic layers at high latitudes (Figure 5d). By contrast, no significance difference was found in the organic content of the CCL layer (mean difference = −0.11 wt%, *z* = −0.86, *p* = 0.83). In addition, significant differences in the organic content of the two layers were detected in the low-latitude population (mean difference = 0.84 wt%, *z* = 6.01, *p <* 0.01), but not in the high-latitude population (mean difference = 0.20 wt%, *z* = 1.49, *p* = 0.44).

## 4. Discussion

We used microsatellites to characterize the population genetic structure of *M. arenaria* across Europe and to test for associations between genetic variables and shell morphological traits. We uncovered evidence for strong population genetic structure, which was unrelated to variation in shell shape, thickness, microstructure and organic content. Instead, although none of our predictor variables explained variation in shell shape, latitude appeared to be the best predictor of variation in shell thickness and organic content. We therefore conclude that most if not all of the observed variation in shell shape and thickness is probably due to environmentally driven phenotypic plasticity.

### 4.1. Population Genetic Structure

Our data are suggestive of the presence of four main genetic groups situated in the Adriatic, the Celtic Sea, on the east coast of Scotland and along the continental coast of the North Sea and the entrance of the Baltic Sea. This observation lends support to a previous study by Cross et al. [27], that documented significant genetic differences between three populations from the British Isles and a single population from the Netherlands. Unfortunately, a direct comparison of the two studies is not possible as we were unable to source samples from the same localities as Cross et al. [27]. This leaves open the question of whether additional genetic groups might be detected around the British Isles as well as in other localities that were not included in the current study, such as the Baltic and White Sea.

Although our sampling design was far from exhaustive, the broad geographical coverage of our study allowed us to capture a number of interesting patterns. First and most obviously, samples from the Adriatic Sea (ITA) were strongly differentiated from the Atlantic populations. Deep genetic divergence between the Mediterranean and the Atlantic is a common pattern found in native European marine invertebrates [71,72], which typically results from a combination of historical vicariance and the cessation of contemporary gene flow due to the presence of the Almería-Oran oceanographic front. However, this is unlikely to provide an explanation in the case of *M. arenaria* for two reasons. First, the soft-shell clam is believed to have gone extinct in Europe during the last glacial maximum [7], which would mean that events pre-dating this period could not have left a genetic trace. Second, after this species was reintroduced into Europe, anthropogenic translocations were commonplace and we are aware of at least two documented instances in which *M. arenaria* was introduced into the Mediterranean [11,12]. Additionally, Lasota et al. [5] suggested that one potential origin of Adriatic *M. arenaria* populations could be the Atlantic coast of North America, which would be consistent with the large magnitude of genetic differentiation observed in the current study. To investigate this further, more extensive geographical sampling would need to be combined with genetic assignment testing in order to evaluate the most likely source of origin(s) of the Mediterranean *M. arenaria* populations. Genomic data would also be desirable as these might allow divergence times to be estimated and the strength of support for alternative recolonization pathways to be evaluated *sensu* [20]. 

The pattern of genetic structure we uncovered among the Atlantic *M. arenaria* populations may superficially resemble that observed in other marine species native to Europe, where Atlantic populations are divided into a northern and a southern genetic cluster [20,73,74]. However, this apparent similarity cannot have originated from the same processes. In the case of marine species native to Europe, this pattern emerged as a consequence of natural postglacial recolonization, either because of the presence of separate glacial refugia in northern and southern Europe or as a result of a founder effect event that occurred during northward recolonization from a single refugium located in southern Europe [74]. By contrast, the recolonization of Europe by *M. arenaria* was at least partially anthropogenic, following its introduction to Denmark between the 13th and 15th centuries [8,9]. This recolonization is known to have proceeded both northward and southward, providing a possible explanation for the overall pattern of isolation by distance in our data.

Finally, we could show that *M. arenaria* samples from Portugal showed high genetic affinity with populations from along the continental coast of the North Sea and the entrance of the Baltic Sea (Le Crotoy (LCT), Balgzand (TEX), Sylt (SYL) and Kiel (KIE)). This finding is again consistent with the notion that man-mediated translocation played an important role in shaping the genetic structure of *M. arenaria* across Europe. In this particular case, the most parsimonious explanation for the observed pattern would be that soft-shell clams from the North Sea coast were introduced into Portugal, either deliberately or inadvertently due to the fact that *M. arenaria* can be easily transported with ballast water [4,5]. In line with the first of these explanations, Conde et al. [13] have already argued that soft-shell clams from Lisbon, as well as from other two Portuguese populations, may have originated as a consequence of intentional introductions.

### 4.2. Shell Shape Variation

An EFA of shell outlines uncovered differences in shell shape among the collection sites, but none of the predictors fitted in our GLMM could explain a significant proportion of the total variation. Although it is not necessarily surprising that the genetic variables were unrelated to variation in shell shape, studies of other bivalve species have reported strong effects of local environmental conditions, substrate type and predation pressure [40,41,44,75]. Consequently, we were somewhat surprised not to have detected any influence of latitude on shell shape in *M. arenaria*. One possible explanation could be that our predictor variables were too crude to capture meaningful variation in key abiotic or biotic variables. In particular, although our broad geographic coverage maximized variation in a suite of climatic and other variables, this may have hindered the detection of relatively subtle effects. This issue could potentially be circumvented by sampling populations over a much finer geographic scale, as this would effectively control for large-scale sources of variability while facilitating more quantitative investigation of specific factors such as substrate typology and sediment size.

### 4.3. Variation in Shell Thickness, Microstructure and Organic Content

Geographic patterns of molluscan skeletal production have typically been explained by two paradigms: poleward reduction of predation pressure [76,77] and increased calcification costs at high latitudes, which results from a combination of decreased CaCO_3_ saturation and reduced metabolic rates [47,48,49,50,51,52,53,54,55,56,57,58,59,60,61,62,63,64,65,66,67,68,69,70,71,72,73,74,75,76,77,78,79]. For *M. arenaria*, predation acts most heavily upon young individuals that are present only in the upper layer of the substrate, while predation risk for older individuals buried deeper in the substrate is negligible [80,81]. Given the size classes analyzed in this study, it could therefore be argued that predation pressure is unlikely to have influenced shell morphology, although we cannot discount the possibility that variation in the risk of predation during early life could have played a role.

Surprisingly, our results are also not consistent with poleward skeletal reductions due to increased calcification costs at high latitudes, as we did not observe any change in total shell thickness along a latitudinal cline. However, shell organic content was reduced in samples from the more northerly, colder environment, which is in line with suggestions that shell organics have higher production costs than CaCO_3_ deposition [46,47]. Similar divergent patterns have been documented for laternulid clams [82], where thicker shells have been suggested to have a selective advantage when individuals are moved through sediment by ice disturbance [83]. Likewise, the calcification pattern we observe might represent a trade-off to preserve a constant shell thickness across latitude. Specifically, the production of less energetically-expensive shell layers may be favored at high latitudes as a means of enhancing protection from physical disturbance.

### 4.4. Genetic Versus Plastic Contributions

Previous studies of a variety of marine and freshwater invertebrates have provided support for a plastic nature of shell morphology. This conclusion has been reached either due to the absence of a link between population genetic structure and morphological variation [30,34,36] or based on the results of reciprocal transplant experiments, which have uncovered a prominent role of environmental variation in shaping shell morphology [84,85]. We built upon the approaches used in previous population genetic studies by integrating genetic variation in the form of principal component scores, which capture multiple aspects of genetic variation and facilitate morphological-genetic comparisons at the level of individuals rather than populations. Furthermore, as heterozygosity is associated with morphological variation in several bivalve species [86,87,88], we included individual sMLH as a predictor in our models. We found no main effects of any of these genetic variables on either shell shape, thickness or organic content, providing evidence for a primary role of phenotypic plasticity, although genomic approaches capable of generating tens of thousands of genetic markers should offer greater power for testing for gene-phenotype associations [36].

## 5. Conclusions

We used microsatellites to characterize the population genetic structure of *M. arenaria* across Europe and to relate this to latitudinal variation in shell shape, thickness, microstructure and organic content. We uncovered strong population structure, consistent with the known involvement of humans in reintroducing this species to Europe as well as a long history of both deliberate and unintentional long-distance translocations. Additionally, we were unable to find any genetic effects on individual variability in shell shape and thickness, consistent with previous studies of other shellfish species reaching the conclusion that shell morphology is largely plastic [34,35,36]. Specifically, the best predictor of the thickness of individual shell layers in *M. arenaria* was latitude, which is associated with variation in numerous variables, both biotic and abiotic.

## Figures and Tables

**Figure 1 genes-11-00298-f001:**
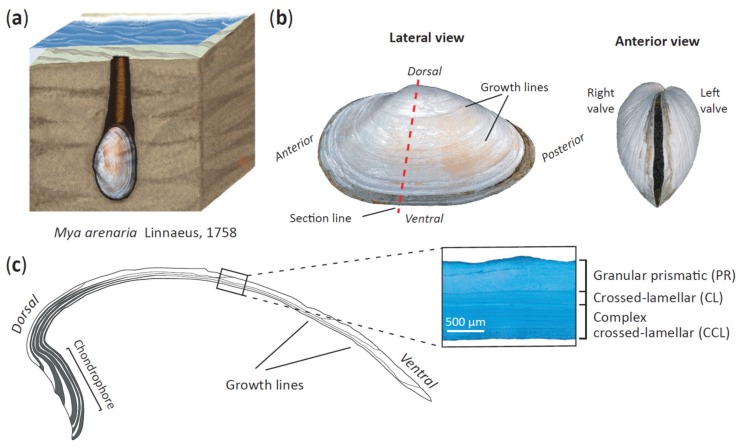
(**a**) Depiction of *Mya arenaria* in its natural habitat; (**b**) lateral and anterior shell views; (**c**) a dorsoventral cross-section of the left shell valve along the axis of maximum growth showing the internal shell structure. Shown are the granular prismatic (PR) layer, the crossed-lamellar (CL) layer and the complex crossed-lamellar (CCL) layer.

**Figure 2 genes-11-00298-f002:**
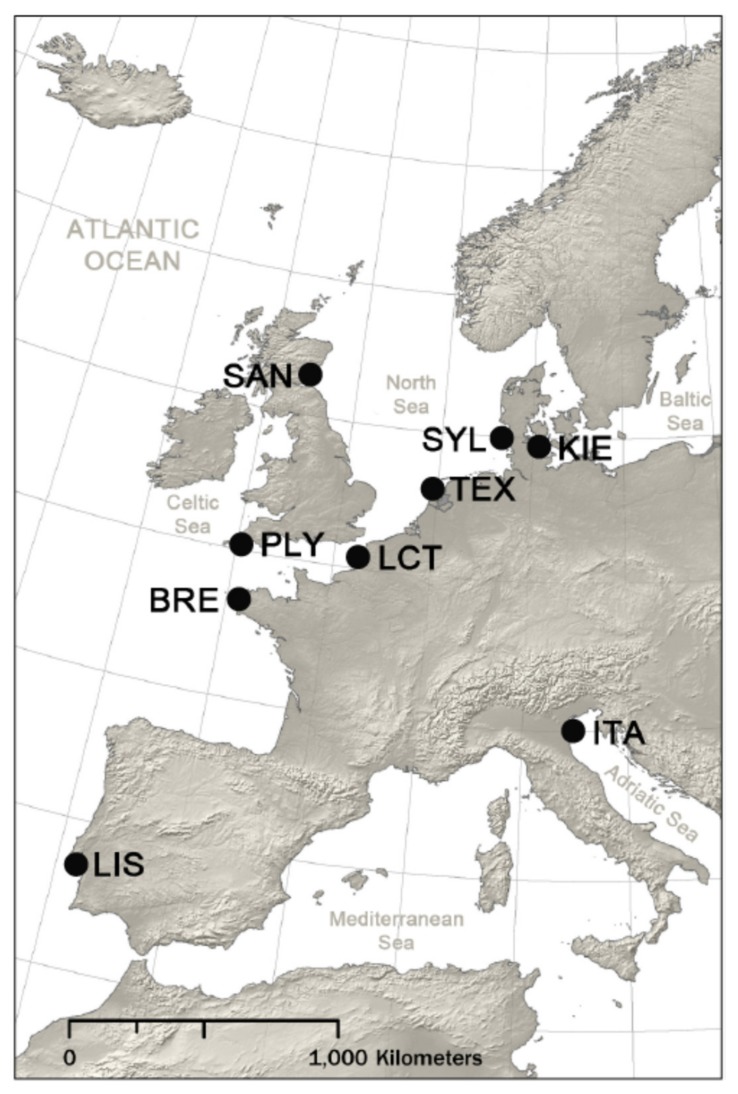
Map showing nine *M. arenaria* sampling locations across Europe: Comacchio, Italy (ITA); Lisbon, Portugal (LIS); Brest, France (BRE); Plymouth, UK (PLY); Saint Andrews, UK (SAN); Le Crotoy, France (LCT); Balgzand, Netherlands (TEX); Sylt, Germany (SYL); Kiel, Germany (KIE)

**Figure 3 genes-11-00298-f003:**
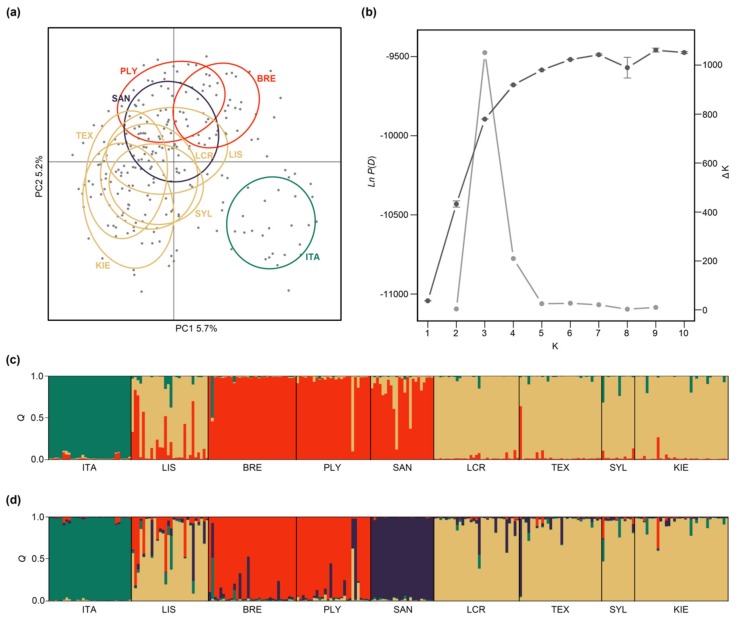
Results of genetic analysis of 247 soft-shell clams genotyped at 11 microsatellites. (**a**) A scatterplot of individual variation in principal component (PC) scores derived from principal component analysis (PCA) of the microsatellite dataset. The amounts of variation explained by each PC are given as percentages and the colored ellipses represent 95% confidence intervals for each population; (**b**) Results of the Structure analysis showing mean and standard deviations of estimated Ln probabilities of the data [*P*(*D*)] (dark grey) and ΔK (light gray) for each value of the number of groups (*K*); Panels (**c**) and (**d**) show estimated group membership coefficients obtained from Structure analyses with the number of groups (*K*) set to three and four respectively. Each individual is represented by a vertical line partitioned into segments of different color, the lengths of which indicate the posterior probability of membership to each group. The populations in panel (**a**) have been color-coded according to the four group solution shown in panel (**d**).

**Figure 4 genes-11-00298-f004:**
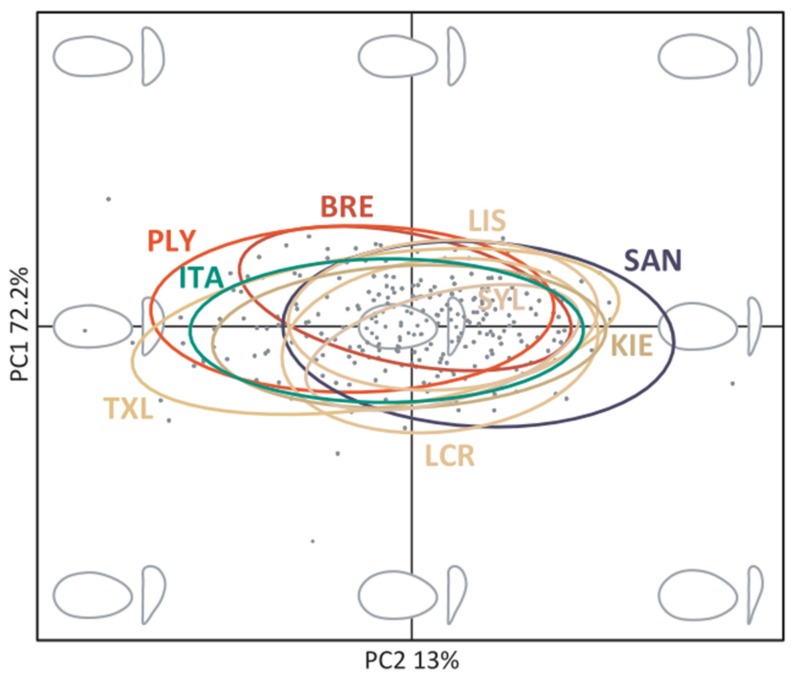
Scatterplot of individual variation in the first two principal components (PCs) from a PCA performed on elliptic Fourier analysis coefficients of lateral and anterior left shell views. The amounts of variation explained by each PC are given as percentages and the ellipses represent 95% confidence intervals for each population. The ellipses are color coded according to the main genetic groups discovered by Structure (shown in Figure 3d). Extreme and average reconstructed shell outlines are shown in grey.

**Figure 5 genes-11-00298-f005:**
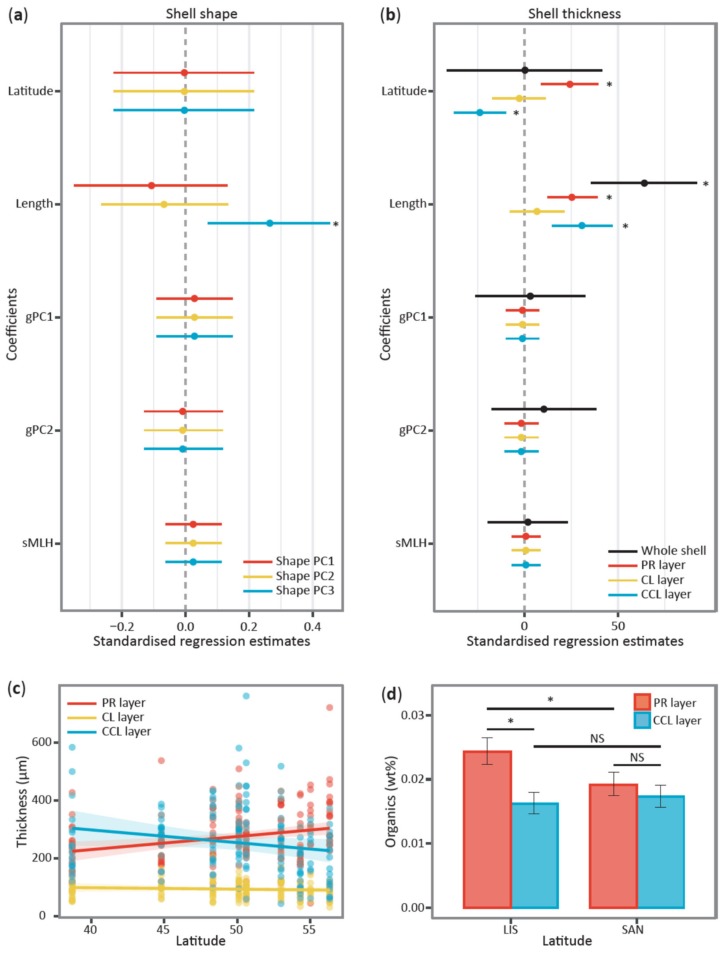
Summary of models of shell shape, thickness and organic content. Panels (**a**) and (**b**) show the mean effect sizes and bootstrapped 95% confidence intervals (CIs) of predictor variables estimated from generalized linear mixed models (GLMMs) of shell shape and shell thickness respectively. Results are summarized in panel (**a**) separately for each of three shape principal components (PCs), which are respectively color-coded in red, yellow and blue respectively. Panel (**b**) summarizes the results of the whole shell thickness model (in black) and the shell layers model, with the granular prismatic (PR) layer shown in red, the crossed-lamellar (CL) layer shown in yellow and the complex crossed-lamellar (CCL) layer shown in blue. Regression parameters were considered statistically significant when the bootstrapped 95% CI (error bars) did not overlap zero (asterisks denote significant differences from zero). (**c**) Relationship between latitude and the thickness of the PR, CL and CCL layers. Mean values (solid lines) and confidence intervals (shaded areas) were predicted while controlling for shell length. (**d**) Latitudinal differences in shell organic content of the PR (red bars) and CCL (blue bars) layers between representative warm temperate (LIS) and cold temperate (SAN) populations. Error bars represent 95% CIs, asterisks represent statistically significant comparisons (*p* < 0.05) and NS denotes non-significant comparisons.

**Table 1 genes-11-00298-t001:** Table of sampling locations, including the number of samples used for the genetic and morphometric analyses. Four genetic diversity statistics are also shown. Observed heterozygosity (H_o_), expected heterozygosity (H_e_), the number of alleles (N_a_) and allelic richness (A_r_) are given as values averaged across loci, with standard deviations reported in parentheses.

Population ID	Location	Samples Used for Genetic Analysis	Samples Used for Shape Analysis	Samples Used for Thickness Analysis	*H* _o_	*H* _e_	*N* _a_	*A* _r_
ITA	Comacchio (Italy)	30	28	18	0.75 (0.13)	0.76 (0.07)	8.23 (2.37)	6.24 (1.6)
LIS	Lisbon (Portugal)	28	29	21	0.67 (0.19)	0.74 (0.18)	8.36 (2.24)	6.56 (1.73)
BRE	Brest (France)	32	35	19	0.76 (0.19)	0.77 (0.12)	9.54 (3.32)	7.08 (2.56)
PLY	Plymouth (UK)	27	30	19	0.66 (0.13)	0.75 (0.09)	8.63 (3.44)	6.6 (2.24)
SAN	Saint Andrews (UK)	23	22	20	0.74 (0.12)	0.72 (0.1)	5.91 (1.64)	5.16 (1.45)
LCR	Le Crotoy (France)	31	40	21	0.71 (0.14)	0.75 (0.08)	8.18 (2.44)	6.36 (1.54)
TXL	Balgzand (Netherlands)	30	30	21	0.71 (0.15)	0.76 (0.1)	8.72 (2.76)	6.36 (1.65)
SYL	Sylt (Germany)	12	13	10	0.62 (0.14)	0.73 (0.13)	6.36 (2.65)	6.63 (2.65)
KIE	Kiel (Germany)	34	35	18	0.59 (0.15)	0.74 (0.11)	8.91 (2.38)	6.29 (1.52)
Total		247	262	167	0.69 (0.06)	0.75 (0.02)	8.09 (1.19)	6.36 (0.52)

**Table 2 genes-11-00298-t002:** Summary of the results of GLMMs of shell shape and shell layer thickness. Estimated statistics and bootstrapped 95% confidence intervals (CIs) for regression parameters are reported for the modelled relationships described in Equations (1) and (2) in the Materials and methods. Because both shell shape and layer were analyzed as categorical variables, shape PC1 and the PR layer were used as the reference levels in the respective models. Regression parameters were considered statistically significant and highlighted in bold when the bootstrapped 95% CI did not overlap zero.

Coefficient	Estimate	SE	95% CI	*t*	*p*-Value
Shell shape GLMM †					
(Intercept)	0.002	0.11	−0.37; 0.37	0.01	0.99
Shape (PC2)	−0.01	0.17	−0.52; 0.49	−0.07	0.94
Shape (PC3)	0.02	0.19	−0.51; 0.53	0.12	0.91
Latitude	−0.004	0.10	−0.23; 0.22	−0.03	0.97
Length × Shape (PC1)	−0.11	0.08	−0.35; 0.13	−1.27	0.20
Length × Shape (PC2)	−0.07	0.10	−0.27; 0.13	−0.70	0.49
Length × Shape (PC3)	0.27	0.10	**0.07; 0.45**	2.70	**0.0072**
gPC1	0.03	0.06	−0.09; 0.15	0.47	0.64
gPC2	−0.01	0.06	−0.13; 0.12	−0.15	0.88
sMLH	0.02	0.04	−0.06; 0.11	0.55	0.58
Shell layers thickness GLMM *					
(Intercept)	274.60	7.15	**257.38; 291.44**	38.41	**<0.0001**
Layer (CL)	−181.29	7.30	**−204.95; −158.30**	−24.82	**<0.0001**
Layer (CCL)	−19.45	14.63	−45.03; 5.82	−1.33	0.18
Latitude × Layer (PR)	24.30	7.36	**9.02; 39.27**	3.30	**0.0010**
Latitude × Layer (CL)	−2.68	3.40	−17.10; 11.27	−0.79	0.43
Latitude × Layer (CCL)	−23.72	13.36	**−37.43; −9.69**	−1.78	**0.076**
Length × Layer (PR)	25.36	6.92	**12.29; 38.93**	3.66	**0.0003**
Length × Layer (CL)	6.74	2.67	−7.63; 21.44	2.53	0.012
Length × Layer (CCL)	30.71	9.79	**14.88; 46.79**	3.14	**0.0018**
gPC1	−0.98	2.65	−9.80; 7.82	−0.37	0.71
gPC2	−1.62	2.45	−10.62; 7.44	−0.66	0.51
sMLH	0.82	2.14	−6.80; 8.64	0.38	0.70

† The random intercepts for the shell shape PCs were normally distributed with mean 0 and variances 0.23, 0.41 and 0.46 (for PC1, PC2 and PC3), respectively. * The random intercepts for the PR, CL and CCL layers were normally distributed with mean 0 and variances 5.78, 3.42 and 30.41, respectively. The variance structure indicates different standard deviations per layer (PR: 1.00; CL: 0.32; CCL: 1.15) and an exponential of the variance covariate gPC2 structure.

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
