# Peer review of "Population Genetic Structure Is Unrelated to Shell Shape, Thickness and Organic Content in European Populations of the Soft-Shell Clam Mya Arenaria"

_genes, 2020, doi:10.3390/genes11030298_

Round 1

Reviewer 1 Report

This manuscript is a very interesting, well-written and thoroughly discussed study of the genetic structure and shell features of soft-shell clam populations in Europe. The study compared 266 samples from 9 sites in Europe, and used microsatellite loci to characterize population structure across locations. The authors carefully interpreted the patterns of genetic structure as resulting from possible human introductions following Pleistocene glaciations. The study also examined whether variation in shell shape, organic content and microstructural elements might be best explained as being related to genotype or to collection site. Generalized linear mixed models (GLMM) were used to explore effects of latitude, size and genotype on shell shape and thickness. The authors described an interesting pattern of shape variation across collection sites, but could not identify the factors driving differences in shell shape. Latitudinal differences in shell microstructure were found.

I have no concerns with the methodology or results; the experimental design is clear and well thought-out, descriptions are very thorough and the results are presented in ample detail in the paper and supplementary materials. The combination of genetic and morphometric approaches helps further our understanding of the factors that can influence shell shape and microstructure in bivalves. This paper will be a useful addition to the literature on molluscan shell shape analysis, and will encourage the use of similar approaches to investigate other species and questions. 

Minor corrections:

Figure 2 legend: please indicate that abbreviations are described in Table 1.

It would be useful to provide the size range of samples used for shape and thickness analysis (possibly in Table 1) - especially since you found a weak relationship between shell shape and shell length.

line 326: I think you mean "which exhibited rounder shells and flatter anterior view profiles".

Fig 5d and line 349: change "SAND" to "SAN"

line 357: "highlighted in bold" - I don't see any bold type in Table 2.

Table 2: What do the superscript symbols represent (for "Shell shape GLMM" and "Shell layers thickness GLMM")?

Author Response

Responses to Review Report (Reviewer 1)

Comments and Suggestions for Authors:

Point 1: This manuscript is a very interesting, well-written and thoroughly discussed study of the genetic structure and shell features of soft-shell clam populations in Europe. The study compared 266 samples from 9 sites in Europe, and used microsatellite loci to characterize population structure across locations. The authors carefully interpreted the patterns of genetic structure as resulting from possible human introductions following Pleistocene glaciations. The study also examined whether variation in shell shape, organic content and microstructural elements might be best explained as being related to genotype or to collection site. Generalized linear mixed models (GLMM) were used to explore effects of latitude, size and genotype on shell shape and thickness. The authors described an interesting pattern of shape variation across collection sites, but could not identify the factors driving differences in shell shape. Latitudinal differences in shell microstructure were found.

I have no concerns with the methodology or results; the experimental design is clear and well thought-out, descriptions are very thorough and the results are presented in ample detail in the paper and supplementary materials. The combination of genetic and morphometric approaches helps further our understanding of the factors that can influence shell shape and microstructure in bivalves. This paper will be a useful addition to the literature on molluscan shell shape analysis, and will encourage the use of similar approaches to investigate other species and questions. 

Response 1: We are glad that the reviewer appreciated our work and that he/she thinks that it represents a useful addition to the literature on molluscan shell shape analysis.

Minor corrections:

Point 2: Figure 2 legend: please indicate that abbreviations are described in Table 1.

Response 2: The legend of Figure 2 now reports that the abbreviations present in the figure are described in Table 1.

Point 3: It would be useful to provide the size range of samples used for shape and thickness analysis (possibly in Table 1) - especially since you found a weak relationship between shell shape and shell length.

Response 3: The number of samples utilized for shape and thickness, as well as genetic, analyses are already provided, separately for each population, in Table 1.

Point 4: line 326: I think you mean "which exhibited rounder shells and flatter anterior view profiles".

Response 4: Yes, that is correct. We have now changed that line according to the reviewer’s comment (See line 354-355 of the revised manuscript with tracked changes). 

Point 5: Fig 5d and line 349: change "SAND" to "SAN"

Response 5: Thank you for noticing that. We have now changed Figure 5 and the mentioned line (See line 378 of the revised manuscript with tracked changes) accordingly. 

Point 6: line 357: "highlighted in bold" - I don't see any bold type in Table 2.

Response 6: We apologize for this. Table 2 in our original manuscript had bold type in it, but this got probably lost in the formatting during submission. We have now paid attention to upload a version of Table 2 containing bold type.  We also include a version in the rebuttal so that the referee can see how this was formatted at the time of submission.

Table 2: Summary of the results of GLMMs of shell shape and shell layer thickness. Estimated statistics and bootstrapped 95% confidence intervals (CIs) for regression parameters are reported for the modelled relationships described in Equations (1) and (2) in the Materials and methods.  Because both shell shape and layer were analysed as categorical variables, shape PC1 and the PR layer were used as the reference levels in the respective models.  Regression parameters were considered statistically significant and highlighted in bold when the bootstrapped 95% CI did not overlap zero.

Coefficient

Estimate

SE

95% CI

t

p-value

Shell shape GLMM†

(Intercept)

0.002

0.11

-0.37: 0.37

0.01

0.99

Shape(PC2)

-0.01

0.17

-0.52; 0.49

-0.07

0.94

Shape(PC3)

0.02

0.19

-0.51: 0.53

0.12

0.91

Latitude

-0.004

0.10

-0.23; 0.22

-0.03

0.97

Length × Shape(PC1)

-0.11

0.08

-0.35; 0.13

-1.27

0.20

Length × Shape(PC2)

-0.07

0.10

-0.27; 0.13

-0.70

0.49

Length × Shape(PC3)

0.27

0.10

0.07: 0.45

2.70

0.0072

gPC1

0.03

0.06

-0.09: 0.15

0.47

0.64

gPC2

-0.01

0.06

-0.13; 0.12

-0.15

0.88

sMLH

0.02

0.04

-0.06: 0.11

0.55

0.58

Shell layers thickness GLMM*

(Intercept)

274.60

7.15

257.38; 291.44

38.41

<.0001

Layer(CL)

-181.29

7.30

-204.95; -158.30

-24.82

<.0001

Layer(CCL)

-19.45

14.63

-45.03; 5.82

-1.33

0.18

Latitude × Layer(PR)

24.30

7.36

9.02; 39.27

3.30

0.0010

Latitude × Layer(CL)

-2.68

3.40

-17.10; 11.27

-0.79

0.43

Latitude × Layer(CCL)

-23.72

13.36

-37.43; -9.69

-1.78

0.076

Length × Layer(PR)

25.36

6.92

12.29; 38.93

3.66

0.0003

Length × Layer(CL)

6.74

2.67

-7.63; 21.44

2.53

0.012

Length × Layer(CCL)

30.71

9.79

14.88; 46.79

3.14

0.0018

gPC1

-0.98

2.65

-9.80; 7.82

-0.37

0.71

gPC2

-1.62

2.45

-10.62; 7.44

-0.66

0.51

sMLH

0.82

2.14

-6.80; 8.64

0.38

0.70

† The random intercepts for the shell shape PCs were normally distributed with

mean 0 and variances 0.23, 0.41, and 0.46 (for PC1, PC2, and PC3), respectively.  

* The random intercepts for the PR, CL and CCL layers were normally distributed with

mean 0 and variances 5.78, 3.42, and 30.40, respectively. The variance structure indicates different standard deviations per layer (PR: 1.00; CL: 0.32; CCL: 1.15) and an exponential of the variance covariate gPC2 structure

Point 7: Table 2: What do the superscript symbols represent (for "Shell shape GLMM" and "Shell layers thickness GLMM")?

Response 7: The superscripts symbols in Table 2 refer to the table’s footnotes. Similarly to the previous comment (see Response 6), Table 2 in our original manuscript contained footnotes, which have probably been lost in the formatting during submission. We have now submitted a version of Table 2 containing such footnotes.  The footnotes are shown in the rebuttal as described in Response 6.

Reviewer 2 Report

Title: Population genetic structure is unrelated to shell shape, thickness
and organic content in European populations of the soft-shell clam Mya
arenaria
Journal: Genes
Authors: Michele De Noia, Luca Telesca, David Vendrami, Hatice Gokalp,
Grégory Charrier, Elizabeth M. Harper, Joseph Hoffman *

Major Issues

The manuscript is well written and very clear.  The authors have sampled several populations of soft-shelled clams from around Europe.  Genetic analysis of this species is lacking, and the paper is unique in this regard.  One weakness that the authors point out is that the species range is very large, and the authors only sample a portion of that range.  Also, these clams were extirpated from this studies sample sites and were recently reintroduced by human activities.  Also, there has been considerable speculation about the genetic component of shell shape and nacre color in bivalves and this is one of the few studies to investigate the connection.

The authors spend considerable time and effort on the morphometric analysis of shell shape.  However, the introduction and background are almost exclusively about population genetics.  This is not reflected in the methods or results section where they spend much of their time discussing shell shape and analysis.  In future versions of the manuscript the intro should be more balanced to reflect what the study is about.

One concern from the GLMM is that shell thickness is strongly correlated with shell length.   This is one of the few significant outcomes of the study.  The authors make the conclusion that latitude changes shell thickness.  However, if longer individuals have thinner shells this indicated rapid growth could this simply be the difference in substrate, average temperatures, or water chemistry at each latitude?  Did the age analysis show that longer/thinner individuals were not older than shorter/thicker individuals?

Further, the discussion is mainly regarding future studies and a desire to include more populations.  Considering the limits of this study and relatively short discussion and conclusions is it possible to reduce the number of morphometric analyses and figures (in particular figure 4 and parts of figure 5, (a) and (c), don’t really show much helpful information)?  Many of these are not significant.  I realize there is already considerable supplementary material for this study, but the conclusions don’t really warrant such a lengthy methods and results section.

Because of the limited sampling the authors can’t make any conclusions regarding population structure other than it exists among these populations.  Are there any other conclusions to be made considering this is a commercially harvested species? 

Author Response

Responses to Review Report (Reviewer 2)

Major Issues

Point 1: The manuscript is well written and very clear.  The authors have sampled several populations of soft-shelled clams from around Europe.  Genetic analysis of this species is lacking, and the paper is unique in this regard.  One weakness that the authors point out is that the species range is very large, and the authors only sample a portion of that range.  Also, these clams were extirpated from this studies sample sites and were recently reintroduced by human activities.  Also, there has been considerable speculation about the genetic component of shell shape and nacre color in bivalves and this is one of the few studies to investigate the connection.

Response 1: We are glad to hear that the reviewer sees our study as unique and that he/she appreciate our investigation regarding the link between genetic component and shell shape. It is correct that we were only able to sample from a limited set of locations, as is the case for most studies.  However, we believe that our coverage is broadly representative and allows us to capture the main patterns that will be present in Europe.

Point 2: The authors spend considerable time and effort on the morphometric analysis of shell shape.  However, the introduction and background are almost exclusively about population genetics.  This is not reflected in the methods or results section where they spend much of their time discussing shell shape and analysis.  In future versions of the manuscript the intro should be more balanced to reflect what the study is about.

Response 2: We agree and have balanced the article by writing a new paragraph on shell morphology for the introduction (See line 79-89 of the revised manuscript with tracked changes).

Point 3: One concern from the GLMM is that shell thickness is strongly correlated with shell length.   This is one of the few significant outcomes of the study.  The authors make the conclusion that latitude changes shell thickness. 

Response 3: The reviewer is correct in saying that shell thickness is correlated with shell length, this is true for both whole shell thickness (i.e.: all shell layers considered together) and for the PR and CCL layers. However, while changes in whole shell thickness are not associated with latitude (see Supplementary Table 4 and Figure 5), the thickness of the PR and CCL layers do show a significant correlation with latitude (see Table 2 and Figure 5). This why we concluded that there is a change in the thickness of the PR and CCL layers with latitude.

Point 4: However, if longer individuals have thinner shells this indicated rapid growth could this simply be the difference in substrate, average temperatures, or water chemistry at each latitude? 

Response 4: We agree with the Reviewer on this point and we have in fact already discussed this topic in our Discussion. Specifically, at line 492-511 we present some of the factors that are likely to affect shell deposition in M. arenaria.

Point 5: Did the age analysis show that longer/thinner individuals were not older than shorter/thicker individuals?

Response 5: We would like to point out that all of our models take into account shell size differences among individuals. Specifically, our models controls for “shell length” in order to avoid any potential bias due to differences in size (and therefore age) among individuals. Our models indicate a positive relationship between shell deposition and length, which is in line with the general pattern of allometric shell growth in bivalves. Therefore, the models show shell thickening with ageing (i.e.: deposition of thicker shell in older individuals) along with change in the relative thickness of internal shell layers depending on latitude.

Point 6: Further, the discussion is mainly regarding future studies and a desire to include more populations.  Considering the limits of this study and relatively short discussion and conclusions is it possible to reduce the number of morphometric analyses and figures (in particular figure 4 and parts of figure 5, (a) and (c), don’t really show much helpful information)?  Many of these are not significant.  I realize there is already considerable supplementary material for this study, but the conclusions don’t really warrant such a lengthy methods and results section.

Response 6: We have taken this point on board and removed one of the morphological panels (Figure 4b).  However, we do not believe that it would be appropriate to remove panels 5a and 5c and associated analyses as they are essential to addressing our study aims.  We furthermore consider that it is poor scientific practice to selectively report only significant effects and we believe these panels do a good job of presenting a combination of both significant and non-significant results in an unbiased manner.

Point 7: Because of the limited sampling the authors can’t make any conclusions regarding population structure other than it exists among these populations.  Are there any other conclusions to be made considering this is a commercially harvested species? 

Response 7: As with all similar studies, we can only reach conclusions based on our sample locations / data.  To our knowledge M. arenaria is not a commercial species in Europe, and we would therefore find it very difficult to discuss genetic structure in the context of commercial harvesting.